# Multi-Grid Capacitive Transducers for Measuring the Surface Profile of Silicon Wafers

**DOI:** 10.3390/mi14010122

**Published:** 2022-12-31

**Authors:** Panpan Zheng, Bingyang Cai, Tao Zhu, Li Yu, Wenjie Wu, Liangcheng Tu

**Affiliations:** 1WuHan National Laboratory for Optoelectronics, Wuhan 430074, China; 2The MOE Key Laboratory of Fundamental, Physical Quantities Measurement, Hubei Key Laboratory of Gravitation and Quantum Physics, PGMF and School of Physics, Huazhong University of Science and Technology, Wuhan 430074, China; 3Beijing Institute of Smart Energy (BISE), Beijing 100085, China; 4TianQin Research Center for Gravitational Physics, School of Physics and Astronomy, Sun Yat-sen University, Zhuhai 519000, China

**Keywords:** surface profile, capacitive transducer, wafer warpage, multi-grid

## Abstract

The measurements of wafers’ surface profile are crucial for safeguarding the fabrication quality of integrated circuits and MEMS devices. The current techniques measure the profile mainly by moving a capacitive or optical spacing sensing probe along multiple lines, which is high-cost and inefficient. This paper presents the calculation, simulation and experiment of a method for measuring the surface profile with arrayed capacitive spacing transducers. The calculation agreed well with the simulation and experiment. Finally, the proposed method was utilized for measuring the profile of a silicon wafer. The result is consistent with that measured by a commercial instrument. As a movement system is not required, the proposed method is promising for industry applications with superior cost and efficiency to the existing technology.

## 1. Introduction

Silicon wafers are widely used in the fabrication process of integrated circuits (IC) and MEMS devices [1,2]. The wafer residual stress is remarkable for the following reasons: The thin pancake-like shape of the wafer, the multiple-layer structure with a different coefficient of thermal expansion (CTE), and the high-temperature fabrication process such as thermal oxidation, hard bake of photoresist, annealing and bonding process [3,4,5,6,7]. The combined effects of the induced residual stresses acting on the backside and frontside of the wafer result in wafer warpage [8]. More recently, larger silicon wafers are always preferred for reducing the cost of a single IC or MEMS device further. Moreover, ultrathin silicon die is a key enabler for high performance semiconductor devices and ultrathin packaging including three-dimensional packages. As a result, measuring the surface profile of the wafers is crucial for safeguarding the precision and reliability of the fabracated IC and MEMS devices [9,10,11,12]. In addition, the measured warpage can also be used for analyzing the residual stress, which is crucial for evaluating the creep property of the fabricated chips. In cases of serious warpage, the wafers will be discarded for saving cost and time in future processing [13].

The main technique for measuring the surface profile of wafers is moving a capacitive or optical spacing sensing probe along multiple lines on the surface of silicon wafers [14,15]; however, the system is complex, high-cost and inefficient. What is more, the complicated multi-axis motorized position system is difficult to integrate with processing equipment for in-suit measurements. Another technique is the optical plane interferometer [16,17,18]. As surface interference has no requirements for moving probes, this method is efficient. However, the measurement precision is limited by the flatness of the reference mirror, which is difficult to fabricate for the large area of the wafer.

In this work, we present a technique for measuring the surface profile of the silicon wafer with arrayed capacitive spacing transducers on a flat glass wafer. The main technology findings include: (1) Through measuring the spacing between the silicon wafer and the glass wafer on multiple grids with arrayed capacitive spacing transducers, the surface profile of the silicon wafer can be fitted without the requirement for an additional movement system. As a result, the system is efficient and easier to integrate into processing machines for in situ measurements. (2) The design of the capacitive transducer with two electrodes on the glass wafer is able to detect the spacing with no need for signal connection to the wafers to be detected, thus simplifying the measurement system. Beneficial from the above innovations, the proposed method is promising for industry applications with superior cost and efficiency to the existing technology.

## 2. Design

A capacitive displacement transducer is composed of two metal electrodes (E1 and E2) deposited on the glass wafer and one electrode formed by the silicon wafer to be measured (E0), as shown in Figure 1.

Based on the planar plate capacitor model, the capacitances can be calculated by:(1)$$C_{1}=C_{2}=\frac{\varepsilon S}{d},$$
where ε is the dielectric constant of air, *S* is the area of the electrodes (*E*_1_ and *E*_2_), and *d* is the spacing between *E*_1_ and *E*_0_. The total capacitance of the two series connected capacitors is:(2)$$C_{12}=\frac{C_{2}}{2}=\frac{\varepsilon S}{2d}.$$

As the capacitance measurement precision (Δ*C*) of the LCR meter is 0.1% of the capacitance to be measured, the precision of the capacitive displacement transducer (Δ*d*) can be evaluated by:(3)$$\Delta d=\frac{2d^{2}}{\varepsilon S}\Delta C=\frac{2d^{2}}{\varepsilon S}\times 0.001\times C_{12}=0.001d.$$

When parasitic capacitance (*C_f_*) is considered, the actual measured capacitance can be expressed as:(4)$$C=C_{f}+\frac{\varepsilon S}{2d}.$$

It can be derived from Equation (4) that the spacing between the *E*_1_ and *E*_0_ is
(5)$$d=\frac{\varepsilon S}{2\left(C-C_{f}\right)}.\;$$

It reveals that the spacing can be calculated via measuring the capacitances *C* and *C_f_*.

In order to measuring the surface profile of the wafer, arrayed electrodes are deposited on a flat glass wafer, as shown in Figure 2. The spacing of every capacitive transducer is calculated according to Equation (5).

The glass wafer was fabricated with typical micromachining processes, as shown in Figure 3. Firstly, a double-layer photoresist was patterned. Then a 40 nm Ti layer and a 200 nm Au layer were deposited with an electron beam evaporation, followed by a lift-off process for patterning the electrodes. Then, another photoresist layer was patterned, covering the pads that are used to connect the LCR meter. Afterwards, a 300 nm SiO_2_ layer was deposited with PECVD for electric insulation. Finally, a wet etching with organic solvent removed the photoresist and the SiO_2_ adhered to it.

The design of the surface profile measurement system is shown in Figure 4. Capacitive transducers are deposited symmetrically on a 5-inch square glass wafer for evaluating the surface profile of 4-inch silicon wafers, as shown in Figure 4a. The capacitive transducer is composed of a circular electrode and a concentric ring with the same area of 28 mm^2^, as shown in Figure 4b. This design is beneficial for reducing the difference between the average distances of the two electrodes to the wafer to be detected. The large electrode size is also beneficial to reduce a fringe effect that is not considered in the model of the capacitive displacement transducer. In addition, arc-shaped marks are utilized for alignment when locating the wafers to be measured.

## 3. Results

### 3.1. Calculation, Simulation and Calibration

The relationship between the measured capacitance and the spacing is calculated according to Equation (2). A finite element analysis (FEA) simulation by Ansoft Maxwell, Ansoft Corporation, Pittsburgh, USA, is used as a reference, as shown in Figure 5. The stray capacitance was simulated in the situation that no wafer is above the electrodes. The results reveal that when correcting the stray capacitance, the simulation results agree well with that of the calculation. It could also be concluded from the results that the capacitive displacement transducer is more sensitive for measuring a smaller spacing. Considering the precision of the LCR meter for measuring capacitances is 0.1%, the precision for measuring the spacing can be calculated based on Equation (3), as shown in Figure 6.

When measuring the surface profile of the wafer, the electrodes are electrically connected to the LCR meter via probes, as shown in Figure 7a. In order to calibrate the capacitive displacement transducer experimentally, a flat silicon chip, whose area is slightly larger than one transducer, was installed on a high-precision displacement table. Firstly, the silicon chip was moved right above the electrodes. Then, it was moved down until contacting the surface of the electrodes. Afterwards, the silicon chip was moved upward by a step of 5 µm, 5 µm, 30 µm, 30 µm, 30 µm, 70 µm, successively. The results are shown in Figure 8 with theoretical value calculated using Equation (4) for reference. The coinciding of the experiment and calculation verified the feasibility of the capacitive spacing transducer. The calibration results were also used to calculate the spacing on every grid when measuring the surface profile of wafers.

### 3.2. Surface Profile Measurements

The warpage of the glass substrate was measured to be less than 0.5 µm by a commercial film-stress interferometer (FLX-2320-S, Toho Technology Corporation, Inazawa, Aichi, Japan), as shown in Figure 9. A 4-inch double-polished silicon wafer with a resistivity of 0.001–0.01 Ω cm was located on the surface profile measurement system, as shown in Figure 8. The capacitance of every spacing transducer was measured by moving the probe to contact the related pads. The measured capacitances were used to calculate the spacing. As a reference, the profile of the wafer was also measured by scanning along four lines on the wafer with a commercial film-stress interferometer. The results are shown in Figure 10. The tendency of the results by the proposed surface profile measurement system agrees well with that by the interferometer. Nevertheless, the errors between them can be several micrometers. The reasons could be: (1) The large area of the electrodes that influences the spatial resolution of every measurement node as the measured spacing is the average spacing of the capacitor. (2) The data of the interferometer were leveled for removing the tilting of the scanned profile, while the data by the proposed capacitive method were not able to be leveled by the same process for different spatial resolution. (3) An error induced by the resolution of the LCR meter, which was less than 0.2 µm theoretically. (4) An error induced by the warpage of the glass substrate, which was less than 0.5 µm.

## 4. Discussion

This paper proposed a surface profile measurement system without the requirement on movement systems, and the proposed method is inexpensive and convenient compared with current techniques. Moreover, the system is easy to be packaged into processing equipment, which is promising to in situ measurements for industry applications.

The proposed surface profile measurement system is only a demo for verifying the feasibility of the design. The errors can be several micrometers, which are not suitable for high-precision applications. In terms of measuring precision, the system can be further improved via the following means: (1) Improving the spatial resolution of the capacitive spacing transducers. For reducing the fringe effect, the dimension of the electrode should be at least three times larger than the spacing [19]. As a result, for measuring a warpage of less than 200 μm, the diameter of the electrode could be designed to be 600 μm, which is 10 times less than the present design. (2) As the dimension of the electrode is reduced, the density of the arrayed capacitive transducers can be increased, which is also beneficial for improving the spatial resolution. (3) Utilizing multichannel capacitance measuring circuits to measure the capacitance of all the capacitive transducers at the same time with higher precision and efficiency. (4) Choosing glass wafer with less original warpage.

## Figures and Tables

**Figure 1 micromachines-14-00122-f001:**
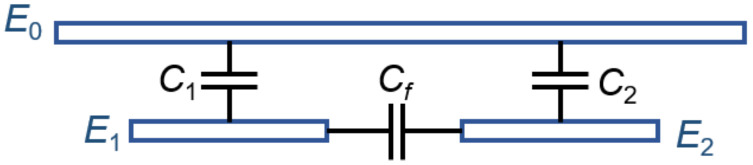
Schematic of a capacitive spacing transducer.

**Figure 2 micromachines-14-00122-f002:**
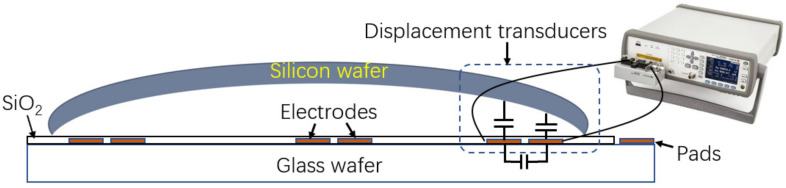
Schematic of multi-grid capacitive transducers for measuring wafer surface profile.

**Figure 3 micromachines-14-00122-f003:**
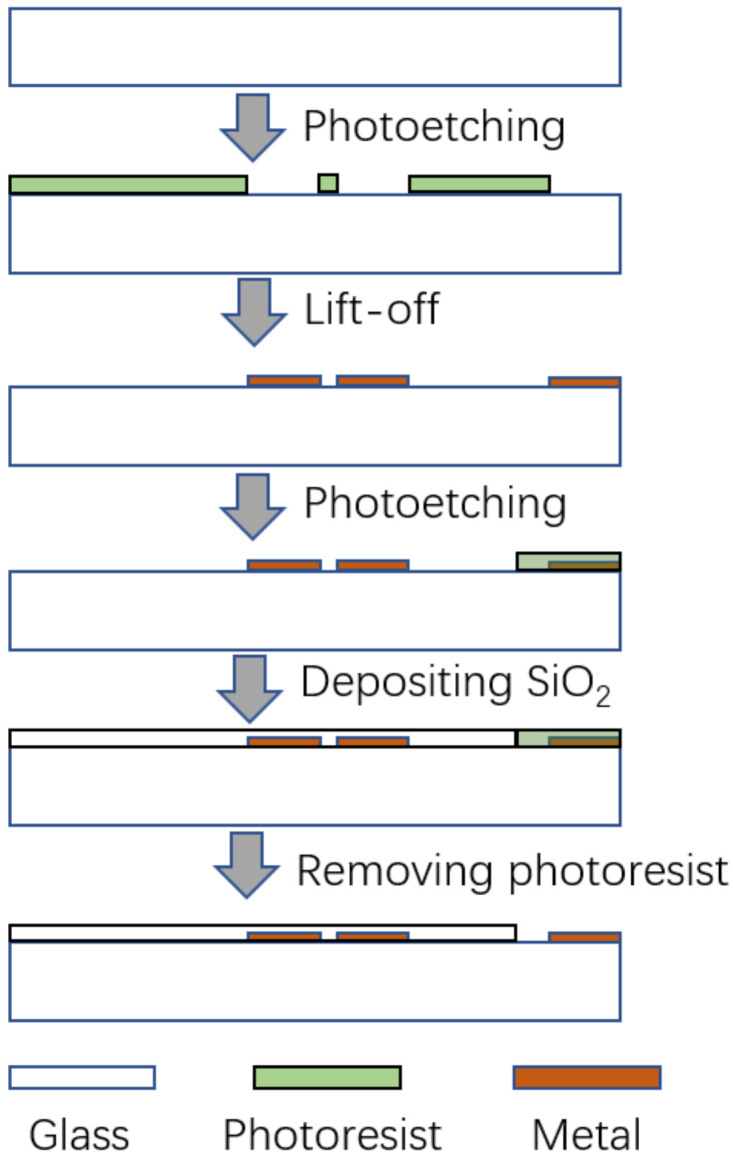
Fabrication process.

**Figure 4 micromachines-14-00122-f004:**
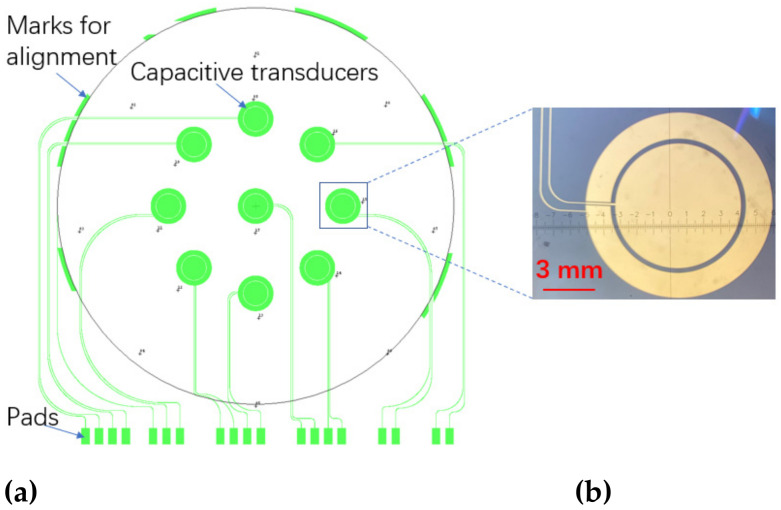
Design of the surface profile measurement system. (**a**) The arrangement of the multi-grid capacitive transducers on a 5-inchsquare glass wafer; (**b**) the microscope image of a single capacitive transducer.

**Figure 5 micromachines-14-00122-f005:**
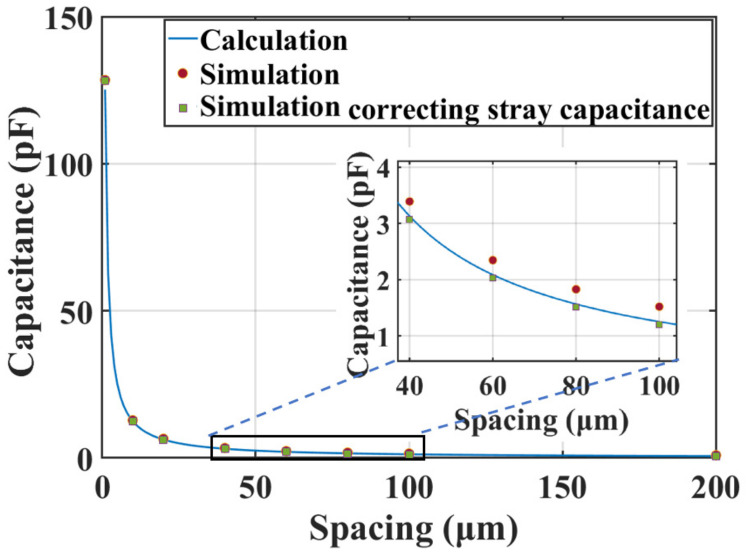
The calculation and simulation of the relationship between the measured capacitance and the spacing.

**Figure 6 micromachines-14-00122-f006:**
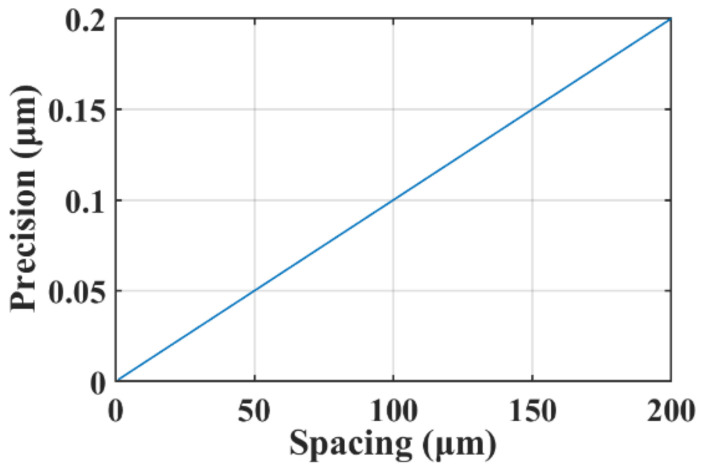
Calculation of the spacing measurement precision.

**Figure 7 micromachines-14-00122-f007:**
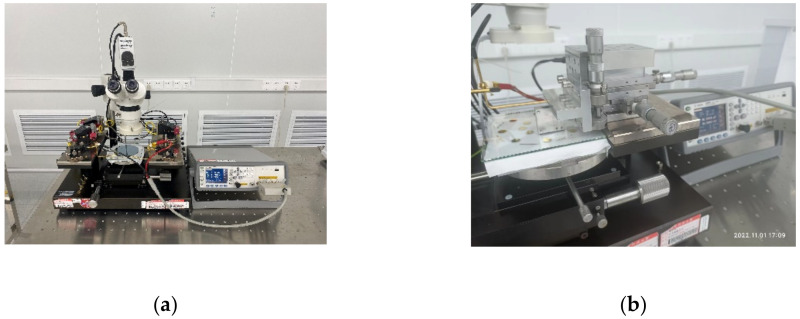
Experimental sets, (**a**) the experimental sets for measuring the surface profile of wafers; (**b**) the experimental sets for calibrating the capacitive spacing transducer.

**Figure 8 micromachines-14-00122-f008:**
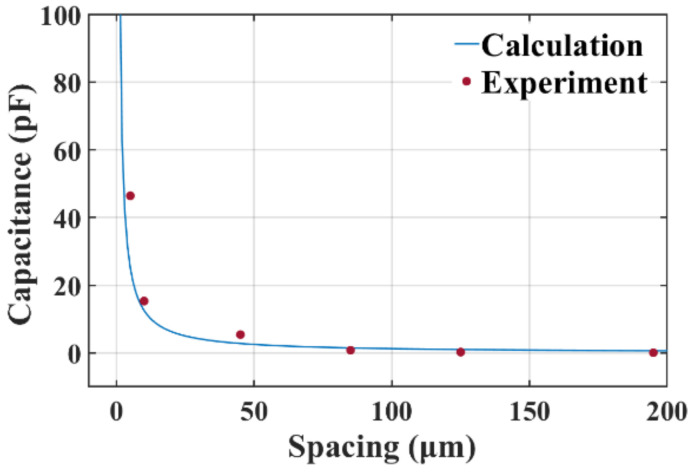
Calibration results of the capacitive spacing transducer.

**Figure 9 micromachines-14-00122-f009:**
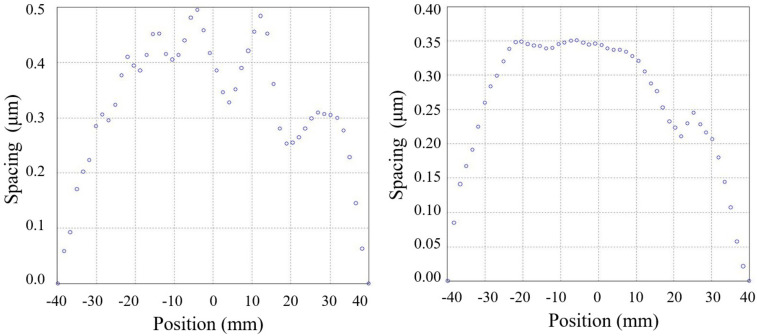
The surface profile of the glass substrate for the capacitive transducers array along orthorhombic directions.

**Figure 10 micromachines-14-00122-f010:**
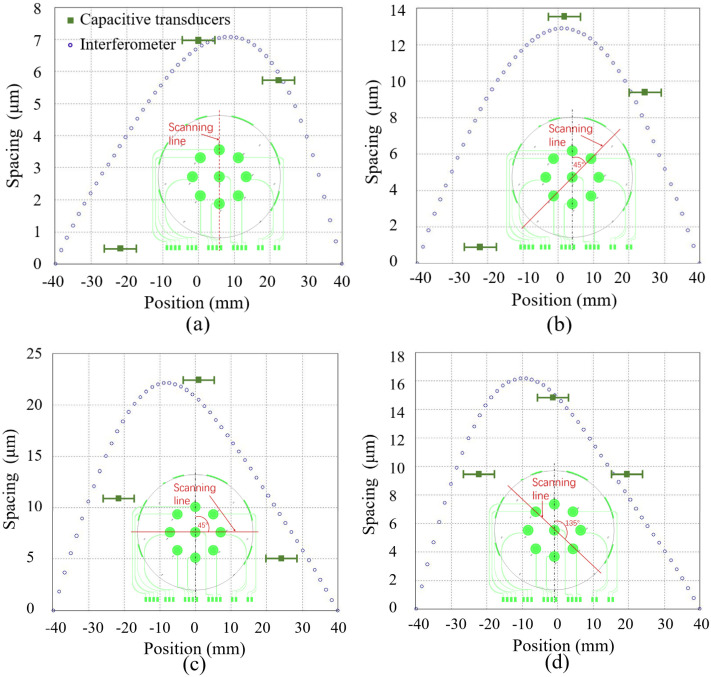
Measured wafer surface profile by the proposed capacitive transducers. The measurement results by a commercial film-stress interferometer are shown as a reference. (**a**) The scanning line is parallel to the reference line; (**b**) The angle between the scanning line and the reference line is 45 degrees; (**c**) The angle between the scanning line and the reference line is 90 degrees; (**d**) The angle between the scanning line and the reference line is 135 degrees.

## Data Availability

Not applicable.

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
