# Peer review of "Multi-Grid Capacitive Transducers for Measuring the Surface Profile of Silicon Wafers"

_micromachines, 2022, doi:10.3390/mi14010122_

Round 1

Reviewer 1 Report

A technique to measure wafer profile using a capacitive transducers array is proposed in this study. The study should be of interest of the readers of this journal. However, revision is necessary prior to that, particularly with regards to the following points.

There is a big gap between the experimental results and the conclusion. A precision of better than 200 nm was mentioned in the conclusion, but there is no enough evidence to support this. Several um error can be clearly observed from Fig. 9. The experimental calibration was also performed in several um steps.

The authors argued about the electrode size which influence the in-plane measurement resolution. Is there any reason to choose the current electrode size? How does the leveling process in the interferometer influence the measurement result? The influence of the warpage of the substrate used for the transducer array should also be elaborated more.

There are many typo in the manuscript. In addition, please confirm the figure pointed in line 114.

Author Response

All the comments by the reviewer are helpful for improving our manuscript. They have been addressed one by one in the revised manuscript. Please see the details in the attached word.

Reviewer 2 Report

2022.11.24

Comments from Reviewer: micromachines-2059415

1.      Recommendation

2.      Comments to Editor:

Title:

Multi-grid Capacitive Transducers for Measuring the Surface Profile of Silicon Wafers Authors:

   Zheng Panpan, Cai Bingyang, Zhu Tao, Yu Li, Wu Wenjie , Tu Liangcheng

General comments:

[Summary of this manuscript]

Surface profile of silicon wafer measurement technique is an extremely important for advanced MEMS device processing such as novel MEMS structure. This manuscript presents very interesting study and informative of surface profile of silicon wafer measurement technique using multi-grid capacitive transducers. Based on these experimental results, the authors concluded that the result is more suitable for designing the surface profile of silicon wafer measurement system with high sensitive and low cost.

Therefore, it is very important to deep understanding of surface profile of silicon wafer measured by multi-grid capacitive transducers for fundamental surface profile of silicon wafer and their silicon semiconductor materials applied engineering point of view.

 [Evaluation of this manuscript]

This manuscript reported that the surface profile of silicon wafer measurement technique is an extremely important for advanced MEMS device processing. This study is an extremely challenging and informative fundamental experimental data for advanced MEMS devices and their structure materials. However, this manuscript has some problems about explanation of introduction and results section. Therefore, the reviewer strongly recommends the authors to reconsider the explanation of introduction and results section.

 Specific comments: 

(1)    What is the technology findings of silicon wafer warpage measurement this manuscript using multi-grid capacitive transducers?

   It is very difficult for reviewer to understand this manuscript of technology findings of silicon wafer warpage measurement for silicon wafer.

  What is author’s study originality and findings using multi-grid capacitive transducers?

  Reviewer strongly recommends improvement of introduction section and additional information.

(2)    Reviewer strongly recommends additional reference (previous study) for improvement of silicon wafer warpage measurement technique some informative research work has already published as follow.

[1] K. H. Yang:” An Optical Imaging Method for Wafer Warpage Measurements”: J. Electrochem. Soc. 132 1214(1985): DOI 10.1149/1.2114066

[2] Michael Raj Marks, Zainuriah Hassan, Member, and Kuan Yew Cheong: “Characterization Methods for Ultrathin Wafer and Die Quality: A Review”, IEEE TRANSACTIONS ON COMPONENTS, PACKAGING AND MANUFACTURING TECHNOLOGY, VOL. 4, NO. 12, DECEMBER 2014; Digital Object Identifier 10.1109/TCPMT.2014.2363570

(3)    Reviewer strongly recommends that the more detail and clear explanation of measurement difference results between capacitive transducers and commercial film-stress interferometer as shown in Figs.9(a),(b) and(c).

(4)    Reviewer strongly recommends that the more detail and clear explanation of “The errors between them are mainly due to the large area of the electrodes that influences the in-plane-position precision. In addition, the leveling process on the data in interferometer also affects the results” as shown in Line136-140.

    Reviewer strongly recommends additional information above the sentence.

(5)    Reviewer strongly recommends that the improvement of manuscript structure (sentence structure) as shown in Line 77.

The glass wafer was fabricated with typical micromachining processes.

[Right and correct sentence]

The glass wafer was fabricated with typical micromachining processes as shown in Fig.3.

(6)    Reviewer strongly recommends that the improvement of manuscript structure (sentence structure) as shown in Line 113-114.

When measuring the surface profile of the wafer, the electrodes are electrically connected to the LCR meter via probes, as shown in Figure 5 (a).

 [Right and correct sentence]

   When measuring the surface profile of the wafer, the electrodes are electrically connected to the LCR meter via probes, as shown in Figure 7 (a).

(7)    Reviewer strongly recommends that the improvement of manuscript structure (sentence structure) as shown in Line 119-120.

The results were shown in Figure 7 with theoretical value calculated using Equation (4) for reference.

[Right and correct sentence]

   The results were shown in Figure 8 with theoretical value calculated using Equation (4) for reference.

(8)    Reviewer strongly recommends that the improvement of manuscript structure (sentence structure) as shown in Line 131-132.

A 4-inch double-polished silicon wafer with a resistivity of 0.001-0.01 Ω cm was located on the surface profile measurement system, as shown in Figure 6 (a).

[Right and correct sentence]

   A 4-inch double-polished silicon wafer with a resistivity of 0.001-0.01 Ω cm was located on the surface profile measurement system, as shown in Figure 9 (a).

Author Response

(The authors gave the same response as above.)

Round 2

Reviewer 1 Report

The authors made some modification to the manuscript in accordance to the review comments. The manuscript is now more suitable for publication. The proposed method has a potential e.g. for in situ warpage monitoring during processes. However, the performance of the proposed method is still limited. More in-depth discussion should be added for further improvement of the proposed method.

The additional discussion regarding the measurement resolution is interesting. So the biggest problem was the spatial resolution, which was defined by the electrode size. However, larger size was preferred to reduce the fringe effect. Can the authors elaborate more regarding how to optimize the electrode size to boost the spatial resolution while limiting the fringe effect?

Please also elaborate more about the glass substrate warpage measurement, did you measure the final fabricated device from the backside? The substrate warpage is currently the 2nd limit after the electrode size. Therefore, its improvement strategy is also interesting to discuss. What was the cause of the warpage? Was the glass substrate bent from the beginning? Or was it due to the SiO2 deposition? In-plane stress could be generated by SiO2 deposition and causes deformation (e.g. 10.1063/1.5045516, 10.1109/JMEMS.2020.2984229). How about the thickness variation of the substrate?

Another small comment, please indicate the scale bar in Fig. 4 (b).

Author Response

The authors appreciate the positive comments by the reviewer. They are all helpful for improving the manuscript. Theses comments have been addressed one by one.

  1. The authors made some modification to the manuscript in accordance to the review comments. The manuscript is now more suitable for publication. The proposed method has a potential e.g. for in situ warpage monitoring during processes. However, the performance of the proposed method is still limited. More in-depth discussion should be added for further improvement of the proposed method.

Response: Thank you for your recommendation. We agree that more in-depth discussion should be added.

Modifications: More in-depth discussion has been added for further improvement of the proposed method at Line 172-183.

  1. The additional discussion regarding the measurement resolution is interesting. So the biggest problem was the spatial resolution, which was defined by the electrode size. However, larger size was preferred to reduce the fringe effect. Can the authors elaborate more regarding how to optimize the electrode size to boost the spatial resolution while limiting the fringe effect?

Response: Thank you for your recommendation.

Modifications: We have elaborated more regarding how to optimize the electrode size to boost the spatial resolution while limiting the fringe effect at Line 175-179.

  1. Please also elaborate more about the glass substrate warpage measurement, did you measure the final fabricated device from the backside? The substrate warpage is currently the 2nd limit after the electrode size. Therefore, its improvement strategy is also interesting to discuss. What was the cause of the warpage? Was the glass substrate bent from the beginning? Or was it due to the SiO2 deposition? In-plane stress could be generated by SiO2 deposition and causes deformation (e.g. 10.1063/1.5045516, 1109/JMEMS.2020.2984229). How about the thickness variation of the substrate?

Response: Thank you for your recommendation. We measured the final fabricated device from the front side. The main cause of the warpage was from the beginning fabrication of the glass substrate, which is verified by experimental measurement. The influence from the SiO2 is weaker may due to: 1) the thickness of the glass substrate is much larger than that of the deposited SiO2. 2) the material of the glass substrate is also SiO2. 3) The process for depositing the SiO2 layer had been optimized with low stress. As we measured the final fabricated device from the front side, the thickness variation of the substrate has no influence on the surface profile of the glass substrate.

Modifications: The main optimization method for reducing the substrate’s warpage was demonstrated at Line 183.

  1. Another small comment, please indicate the scale bar in Fig. 4 (b).

Response: Thank you for your recommendation.

Modifications: The scale bar was indicated in Fig. 4 (b).

Reviewer 2 Report

General comments:

[Summary of this manuscript]

I read above the revised manuscript from authors carefully.

I understand that the authors can clearly answer the specific comments from reviewer.

Therefore, my opinion is accepted this revised manuscript by Micromachines.

Author Response

The authors appreciate the positive comments by the reviewer. 

Round 3

Reviewer 1 Report

The authors have provided sufficient improvement to the manuscript.